DNA barcoding for biodiversity assessment: Croatian stoneflies (Insecta: Plecoptera)

Hlebec Dora 1 2 3 dora.hlebec@biol.pmf.hr
Sivec Ignac 4
Podnar Martina 5
Kučinić Mladen 1
1 Department of Biology, Faculty of Science, University of Zagreb , Zagreb , Croatia
2 Zoological Museum Hamburg, Leibniz Institute for the Analysis of Biodiversity Change , Hamburg , Germany
3 Croatian Biospeleological Society , Zagreb , Croatia
4 Slovenian Museum of Natural History , Ljubljana , Slovenia
5 Croatian Natural History Museum , Zagreb , Croatia
Tang Xiaotian
Electronic publication date: 2022 Apr 20
Publication date: 2022
Volume: 10
Electronic Location ID: e13213
Received 2021 Dec 27; Accepted 2022 Mar 12
Copyright: © 2022 Hlebec et al.
Copyright year: 2022
Copyright holder: Hlebec et al.
License: This is an open access article distributed under the terms of the Creative Commons Attribution License, which permits unrestricted use, distribution, reproduction and adaptation in any medium and for any purpose provided that it is properly attributed. For attribution, the original author(s), title, publication source (PeerJ) and either DOI or URL of the article must be cited.
License URL: https://creativecommons.org/licenses/by/4.0/

Keywords: DNA barcode library, Plecoptera, Morphology, Species delimitation, Bioindicators, COI, Mitochondrial DNA, Water quality

Funding: Croatian Science Foundation IP-2016-06-9988 Dora Hlebec through ESF DOK-2018-09-1417 This research was funded by the Croatian Science Foundation (project DNA barcoding of Croatian faunal biodiversity IP-2016-06-9988) and Dora Hlebec through ESF (DOK-2018-09-1417). The funders had no role in study design, data collection and analysis, decision to publish, or preparation of the manuscript.

==============================
Background

The hemi-metabolous aquatic order Plecoptera (stoneflies) constitutes an indispensable part of terrestrial and aquatic food webs due to their specific life cycle and habitat requirements. Stoneflies are considered one of the most sensitive groups to environmental changes in freshwater ecosystems and anthropogenic changes have caused range contraction of many species. Given the critical threat to stoneflies, the study of their distribution, morphological variability and genetic diversity should be one of the priorities in conservation biology. However, some aspects about stoneflies, especially a fully resolved phylogeny and their patterns of distribution are not well known. A study that includes comprehensive field research and combines morphological and molecular identification of stoneflies has not been conducted in Croatia so far. Thus, the major aim of this study was to regenerate a comprehensive and taxonomically well-curated DNA barcode database for Croatian stoneflies, to highlight the morphological variability obtained for several species and to elucidate results in light of recent taxonomy.

Methods

A morphological examination of adult specimens was made using basic characteristics for distinguishing species: terminalia in males and females, head and pronotum patterns, penial morphology, and egg structures. DNA barcoding was applied to many specimens to help circumscribe known species, identify cryptic or yet undescribed species, and to construct a preliminary phylogeny for Croatian stoneflies.

Results

Sequences (658 bp in length) of 74 morphospecies from all families present in Croatia were recovered from 87% of the analysed specimens (355 of 410), with one partial sequence of 605 bp in length for Capnopsis schilleri balcanica Zwick, 1984. A total of 84% morphological species could be unambiguously identified using COI sequences. Species delineation methods confirmed the existence of five deeply divergent genetic lineages, with monophyletic origin, which also differ morphologically from their congeners and represent distinct entities. BIN (Barcode Index Number) assignment and species delineation methods clustered COI sequences into different numbers of operational taxonomic units (OTUs). ASAP delimited 76 putative species and achieved a maximum match score with morphology (97%). ABGD resulted in 62 and mPTP in 61 OTUs, indicating a more conservative approach. Most BINs were congruent with traditionally recognized species. Deep intraspecific genetic divergences in some clades highlighted the need for taxonomic revision in several species-complexes and species-groups. Research has yielded the first molecular characterization of nine species, with most having restricted distributions and confirmed the existence of several species which had been declared extinct regionally.

Introduction

With 3,800 described species across 17 families (Fochetti & Tierno de Figueroa, 2008; DeWalt et al., 2021; South et al., 2021), the ancient (Béthoux et al., 2011) hemi-metabolous insect order Plecoptera, commonly known as stoneflies, represents an important component of freshwater ecological systems and terrestrial and aquatic food webs (Fochetti & Tierno de Figueroa, 2008; South et al., 2020). Stoneflies are widely distributed on all continents except Antarctica (Zwick, 2000) and their range and abundance have declined rapidly in the last 30 years in Central Europe (Fochetti & Tierno de Figueroa, 2006; Bojková et al., 2012), mainly due to anthropogenic influences (i.e., habitat destruction and pollution) (Fochetti & Tierno de Figueroa, 2008; Bálint et al., 2011). Typical stonefly habitat is a lotic system characterized by cold, fast flowing and well-oxygenated water (Sivec & Yule, 2004) in the mountains of temperate regions. Typically, stoneflies are macropterous or brachypterous, but several species, especially males at higher elevations are apterous or micropterous (Illies, 1966). Due to their poor dispersal capacity (Lee et al., 2022), stoneflies are ideal organisms for biogeographical studies (Murányi, 2011; Graf et al., 2012; Pessino et al., 2014; Stevens, Bishop & Picker, 2018; Gamboa et al., 2019).

Since Plecoptera taxa exhibit high degrees of endemism and morphological diversity (Fochetti & Tierno de Figueroa, 2006; Murányi, 2011; Murányi, Kovács & Orci, 2016), regional field research has both local and global value. Prior to 2009 when concerted efforts in field research began, only 28 species of stoneflies were recorded in Croatia (Sivec, 1980, 1985), a surprisingly low number considering the many suitable habitats (Illies, 1978) and compared to checklists for neighbouring countries: Slovenia with more than 100 species (Sivec, 2001), Bosnia and Herzegovina with 73 species and subspecies (Kaćanski, 1976), Montenegro with 57 species (Murányi, 2008), Hungary with 61 species (Andrikovics & Murányi, 2001) and Serbia with 90 documented species (Petrović et al., 2014). Greater efforts in Croatia resulted in 50 species (Popijač, 2008; Popijač & Sivec, 2009a), but still studies were limited to narrow areas such as Plitvice Lakes National Park (Popijač & Sivec, 2009b; Ridl et al., 2018), Cetina River (Popijač & Sivec, 2009b), Čabranka and Gerovčica Rivers (Popijač & Sivec, 2009c) and lower reaches of the Una River and its tributaries (Popijač & Sivec, 2011). In Plitvice Lakes National Park during the above-mentioned studies, some specimens were recorded that could not be assigned with certainty to known species: Perlodes cf. intricatus, (Pictet, 1841) Isoperla cf. lugens (Klapálek, 1923), Leuctra cf. pusilla Krno, 1985, Leuctra sp., Nemoura sp., Protonemura sp., Isoperla sp. and Perlodes sp. Also, during these studies, several remarkable species were documented: Marthamea vitripennis Burmeister, 1839, which was re-discovered again after one century (Sivec, 1985; Popijač & Sivec, 2011), Perla burmeisteriana Claassen, 1936 (Popijač & Sivec, 2009a), Besdolus imhoffi Pictet, 1841 (Popijač & Sivec, 2010) and Protonemura julia Nicolai, 1983 (Popijač & Sivec, 2009c).

Currently, the most widely accepted system of stonefly classification is by Zwick (2000), with two recognized suborders: Arctoperlaria and Antarctoperlaria. To resolve deeper phylogenetic relationships, research has highlighted the need for molecular data, which in the last years, at least in part, has helped overcome morphology-based identification limitations. South et al. (2020) produced the most complete molecular phylogenetic study of stoneflies based on the North American fauna, following several studies with limited taxon sampling (Thomas et al., 2000; Chen et al., 2018; Wang et al., 2018; Ding et al., 2019). Allopatric diversification, driven mostly by glaciation and orogenesis, has been the main contributor to current diversity patterns in Plecoptera (Zwick, 2000; Weiss, Stradner & Graf, 2011; Theissinger et al., 2013).

DNA barcoding, which uses sequence diversity in the standardized 658-bp region of the mitochondrial cytochrome c oxidase subunit I (COI) gene to aid in species identification and circumscription, is a useful tool for associating immature life stages and to identify cryptic species. Such an approach is extremely important for biodiversity assessment (Hebert et al., 2003a, 2004; Hebert, Ratnasingham & deWaard, 2003b; Valentini, Pompanon & Taberlet, 2009; Morinière et al., 2017). The application of DNA barcoding has aided species delimitation in various groups of organisms and has pointed to divergent haplotypes and hybridization (Van Velzen et al., 2012; Szivák et al., 2017; Zangl et al., 2019, 2021). Once a set of barcodes for a group of organisms is established, examination of previously unidentified specimens is greatly facilitated (DeSalle, 2006; Ratnasingham & Hebert, 2007). However, it has been observed that the results of DNA barcoding could be confounded (Havemann et al., 2018) due to Wolbachia infections (Werren, Zhang & Guo, 1995), incomplete lineage sorting (Petit & Excoffier, 2009), pseudogenes (Ribeiro Leite, 2012), introgressive hybridization and recent speciation (Raupach et al., 2014). DNA barcoding has proved to be a great tool for identification of species from the EPT (Ephemeroptera, Plecoptera and Trichoptera) group (Gill et al., 2014; Ball et al., 2005; Webb et al., 2012; Morinière et al., 2017; Kučinić et al., 2020). Furthermore, the analysis of DNA barcoding results is often difficult due to deposited barcode sequences without scientific species names, known as “dark taxa”, which represent groups of organisms characterized with a lack of taxonomic expertise or undescribed species (Page, 2016; Ryberg & Nilsson, 2018). Additionally, results from DNA barcoding and the traditional taxonomic approach can be incongruent and this is not surprising given the process of evolution itself (Hendrich et al., 2010, 2014). Therefore, for efficient species delimitation, it is necessary to analyze multiple character systems and use integrative taxonomy (Vitecek et al., 2017a; Zhang et al., 2013).

Conducted in the framework of the project DNA barcoding of Croatian faunal biodiversity, the present study aims at: (i) developing a DNA reference barcode library for the Croatian stonefly fauna with macrophotographs of 26 species, (ii) getting first insight into inter- and intraspecific genetic diversity, (iii) assessing morphological variability of stoneflies in Croatia, (iv) highlighting localities with high biodiversity, especially in isolated habitats of the Dinaric Karst area to assist conservation planning and strategies for protecting the genetic diversity of stoneflies, and (v) filling the gaps in the Barcode of Life Data System database (BOLD). Furthermore, the study will contribute to knowledge about species distribution, species complexes and systematic and phylogenetic relationships.

Materials and Methods

Taxon sampling

Specimen collection was conducted with approval from the Ministry of Economy and Sustainable Development of the Republic of Croatia (UP/I-612-07/21-48/73). A total of 337 stonefly specimens (Data S1, Table S1) from 95 different localities in Croatia and eighteen specimens from seventeen localities in Slovenia (Fig. S1) were used in phylogenetic analysis. Information regarding species determinations and details about sampling sites can be found in Table S1. Adult specimens were collected using sweep nets and beating sheets, while larval specimens were collected by handpicking. The aedeagus of males was everted in the field and specimens were fixed and stored in 96% ethanol for morphological and molecular analysis. Morphological characteristics of male terminalia were examined after KOH treatment (Stark & Gaufin, 1978).

Comparative study on morphology was made using specimens kept in the Collection of stoneflies in the Slovenian Museum of Natural History, Ljubljana, Slovenia (PMSL). Newly collected specimens are deposited in the Croatian Natural History Museum, Zagreb, Croatia (CNHM), under the Collection of Plecoptera Sivec & Hlebec (CPSH). Voucher information for individuals used in phylogenetic analysis are publicly accessible in BOLD (dx.doi.org/10.5883/DS-CROPL) and GenBank under the accession numbers as listed in Table S1.

Individuals were sorted and identified using a Leica Wild M3Z stereomicroscope. Macrophotographs were taken using a Canon EOS 5D Mark II. Morphological examination was made using descriptions and identification keys: Illies, 1955; Kaćanski & Zwick, 1970; Raušer, 1980; Ravizza, 2002; Sivec & Stark, 2002; Graf & Schmidt-Kloiber, 2003; Zwick, 2004; Murányi, 2011. The most reliable diagnostic characters for species determination were terminalia in males and females, head and pronotum patterns, penial armatures and egg structures for species within the genus Perla.

DNA extraction, gene amplification and sequencing

The number of specimens per species for COI marker amplification ranged from one to 24 (Isoperla inermis Kaćanski & Zwick, 1970; see Table S1). Tissue was dissected from a single leg for each specimen and genomic DNA was extracted from recently collected samples using the Sigma GenElute Mammalian Genomic DNA Miniprep Kit (Sigma-Aldrich, St. Louis, MI, USA) and from specimens older than 10 years using the QIAamp DNA Micro Kit (Qiagen, Hilden, Germany) and eluting in 50 µl of elution buffer to increase DNA yield. A partial region of the mitochondrial cytochrome c oxidase subunit I gene (COI, the DNA barcode region, Hebert et al., 2003a) was amplified using two sets of primers: (a) LCO-1490/HCO-2198 (Folmer et al., 1994) or (b) C_LepFolF/C_LepFolR (Folmer et al., 1994; Hebert, Ratnasingham & deWaard, 2003b). For samples older than 10 years, the DNA barcode region was amplified as shorter overlapping fragments with primer set: (c) MLepF1/LepR1 and MLepR1/LepF1 (Hajibabaei et al., 2006). All polymerase chain reactions (PCRs) were carried out in a total volume of 20 µl with the PCR mixture prepared following Hlebec et al. (2021). Thermocycling conditions are given in Table S2. PCR products were purified using Exonuclease I (0.05 U/µL) and FastAP (0.025 U/µL) enzymatic system (Thermo Fisher Scientific, Inc., Waltham, MA, USA). The reaction was carried out using the following conditions: 37 °C for 1 hr followed by 80 °C for 20 min. Bi-directional sequencing was done by Macrogen Inc. (Amsterdam, Netherlands), using amplification primers.

Sequence editing and phylogenetic analyses

In total, 355 obtained sequences were checked and inspected manually for base pair ambiguities, indels and stop codons in Geneious Prime 2021.2 (Biomatters, Auckland, New Zealand) to confirm overall sequences quality. Sequences were aligned using MAFFT ver. 7 (Katoh & Standley, 2013). The final alignment for the COI gene fragment was 658 bp in length (Data S2). Sequences were collapsed into 268 COI haplotypes using the online tool FaBox ver. 1.5 (Villesen, 2007). Evolutionary divergence was estimated using uncorrected pairwise genetic distances (p-distances) in MEGA-X ver. 10.2.6 (Kumar et al., 2018) (Figs. 1A–1D, mean values are shown in Table S3). Phylogenetic analyses were performed using Maximum likelihood (ML) optimality criteria in IQ-TREE2 (Minh et al., 2020) under a GTR + I + G optimal model of nucleotide evolution (as determined by jModelTest ver. 2.1.10 (Darriba et al., 2012) under the Bayesian information criterion (BIC)) and bootstrapping with 2,000 ultrafast bootstrap replicates (Hoang et al., 2018) (Fig. 2). ABMAY005-09 (Heptageniidae) and ABMAY015-09 (Stenacron interpunctatum Say, 1839) were selected as outgroups. Additionally, phylogenetic relationships between haplotypes were inferred using the Neighbour joining (NJ) method in MEGA-X ver. 10.2.6 (Kumar et al., 2018) (Fig. S2) and Bayesian inference (BI) in MrBayes ver. 3.2.7 (Ronquist et al., 2012) via the online CIPRES Science Gateway ver. 3.3 (Miller, Pfeiffer & Schwartz, 2010) (Fig. S3). The NJ analysis was performed using the Kimura-2-parameter (K2P) model with the pairwise deletion option. Bootstrap support was inferred using the fast bootstrap algorithm, based on 5,000 replicates. For the BI, two separate runs with four Markov chain Monte Carlo (MCMC) simulations were performed for 10 million generations, sampling every 1,000 generations, using ABMAY005-09 (Heptageniidae) as the root and discarding the initial 25% of the trees as burn-in. Remaining trees were used to create a 50% majority rule consensus tree, with nodal values representing the posterior probabilities. TRACER ver. 1.7.1 (Rambaut et al., 2018) was used to check the convergence between the two runs. Phylogenetic trees were visualized using FigTree ver. 1.4.3 (Rambaut, 2009) and iTOL ver. 5 (Letunic & Bork, 2021). Existence of a barcoding gap (distance between the mean intraspecific sequence variability and interspecific variability for congeneric COI sequences) was ascertained using the “Barcode Gap Analysis” tool, provided in BOLD, using the Kimura-2-Parameter (K2P) distance metric (Puillandre et al., 2012b) (Figs. 1A–1D, Table S4). To visualize phylogeographic relationships among specimens from the largest observed lineage, a median joining (MJ) network (Bandelt, Forster & Röhl, 1999) among 12 haplotypes (Fig. 3, Data S3) was generated using the program PopART ver. 1.7 (Leigh & Bryant, 2015) with default settings. Six different alignments were created for additional phylogenetic (ML in IQ-TREE2 with settings as above, under the optimal model of nucleotide evolution as is listed in Table S5) and species delineation analysis in order to interpret the results and to check the plausibility. We applied these analyses to: (a) species with high levels of intraspecific morphological variability (Perlodes intricatus (Pictet, 1841) and Isoperla inermis), (b) specimens that differed from the described morphospecies (Isoperla cf. lugens, Protonemura hrabei (Raušer, 1956) and Taeniopteryx n.sp. CRO-1) and (c) species, which were taxonomically interesting (Besdolus imhoffi (Pictet, 1841) and B. illyricus Kovács & Zwick, 2008). Analyses were performed with sequences of these species and their closely related congeners from BOLD (Ratnasingham & Hebert, 2007, http://www.boldsystems.org) and GenBank (https://www.ncbi.nlm.nih.gov/) databases (accessed 20 August 2021). Accession numbers of all sequences used in additional analysis are listed in Datasets 1–6, Table S5. Molecular species delineation was achieved through four different methods: the BIN (Barcode Index Number) assignment tool on the BOLD server using the refined single linkage (RESL) algorithm (Ratnasingham & Hebert, 2013), ABGD (Automatic Barcode Gap Discovery) (Puillandre et al., 2012a), ASAP (Assemble Species by Automatic Partitioning) (Puillandre, Brouillet & Achaz, 2021) and mPTP (Multi-rate Poisson Tree Processes) (Kapli et al., 2017). All methods clustered COI sequences into Operational Taxonomic Units (OTUs) based on sequence similarity.

Figure 1 Box plot of uncorrected pairwise genetic distances (p-distances) (A) and results of “Barcode Gap Analysis” (B–D).

(A) Sorted by distance category: intraspecific (specimens that belong to the same species), congeneric (specimens belonging to different species, but to the same genus) and confamiliar (specimens that belong to the same family). Boxes indicate interquartile range (IQR: between upper (Q3) and lower (Q1) quartile). Black bars designate medians, whiskers indicate values within 1.5 × IQR beneath Q1 or 1.5 × above Q3. Circles depict outliers (above or below 1.5 × IQR). (B) The barcode gap for 74 species of Croatian stoneflies shown by plotting maximum intraspecific distance against interspecific (nearest-neighbour) distance. Dots above the diagonal indicate species with a barcode gap. (C) Scatterplot plots the mean intraspecific distances against the minimum interspecific distances. (D) Scatterplot plots the number of individuals in each species against their maximum intraspecific distances.

Figure 2 Circular maximum-likelihood (ML) phylogram from analysis of the released dataset and results of species delineation methods.

Maximum likelihood phylogeny based on the DNA barcoding region (5′ fragment of the mitochondrial COI gene). Species are colour-coded by family. Dots on nodes represent ultrafast bootstrap support values (BS) categories and the size of dots is proportional with the value. The results of species delineations are represented with the bars in different colours and indicate the OTUs inferred by ABGD, ASAP and mPTP methods. Terminal codes present BOLD IDs, as in Table S1. An asterisk indicates two tentative species within Isoperla inermis specimens inferred by ASAP method. The tree was annotated in FigTree ver. 1.4.3 (Rambaut, 2009) and iTOL ver. 5 (Letunic & Bork, 2021) and finished in Adobe Illustrator.

Figure 3 Sampling sites and median-joining network of 658 bp long Isoperla cf. lugens COI sequences.

(A) Map of Croatia and neighbouring countries with sampling localities (colour coding matches insert in 3B). (B) MJ network of COI sequences. Colours and numbers indicate different sampling localities. Numbers of mutational steps are given as hatch marks. The black dots indicate the extinct ancestral or unsampled haplotypes. Frequencies of the haplotypes are proportional to the size of the circles. Haplotypes are labelled with BOLD IDs, as in Table S1. Approximate border of the Dinaric Karst according to Gams (2004). Map is produced with Cartopy package 0.19 in Python with use of elevation data from European Union, Copernicus Land Monitoring Service (2016). In order to distinguish similar colours we included additional number coding.

Members of a BIN usually belong to a species recognized using traditional morphological analysis and taxonomy (Hendrich et al., 2014) and species assignment is based on a universal upper threshold for intraspecific distances (e.g., 2.2%) (Ratnasingham & Hebert, 2013). BIN counts are usually used for species richness (Hebert et al., 2016), but single-locus delineation methods tend to oversplit by mistaking different lineages within populations as putative species (Muster & Michalik, 2020; Meier et al., 2021). Use of the “BIN Discordance” tool on BOLD, provides insight into the concordance between barcode sequence clusters and species designations.

Automatic Barcode Gap Discovery (ABGD) was carried out on the web server (https://bioinfo.mnhn.fr/abi/public/abgd/abgdweb.html), applying the K2P model and using default parameters, except for the relative gap width, which was set to X = 1.0. ASAP was also carried out on the web server (https://bioinfo.mnhn.fr/abi/public/asap/asapweb.html) using p-distances with default settings. The mPTP method, run on the web server (http://mptp.h-its.org/), was implemented using default settings and inputting the ML tree from IQ-TREE2 (Minh et al., 2020), as the starting tree.

Validity and reliability of the generated DNA barcode library was evaluated by comparing classical taxonomy with the counts of OTUs from the various species delineation methods.

Results

Field sampling and morphological identification produced 74 species of stoneflies in seven families and nineteen genera. Four species (Capnopsis schilleri balcanica, Zwicknia rupprechti Murányi, Orci & Gamboa, 2014, Zwicknia bifrons (Newman, 1838) and Nemoura sciurus Aubert, 1949) were identified by comparing with available sequences in the BOLD and GenBank databases.

Barcode sequencing was successful for 355 of 410 (87%) individuals. All COI sequences were 658 bp in length, except for one sequence of Capnopsis schilleri balcanica, which was 605 bp in length.

Molecular results were largely congruent with morphological assessment such that 62 of the 74 morphological species (84%) could be unambiguously identified using COI sequences alone. The median number of barcodes per species was four, and seventeen predominantly rare species were known only from single specimens.

The average confamilial p-distance was 20.8% (ranging from 18.2–23.4%) while the average congeneric distance was 15.6% (ranging from 12.1–19.2%). The mean intraspecific p-distance was 1.3% (ranging from 0–6.8%) (Fig. 1A). The maximum intraspecific distance of 6.8% was obtained for Isoperla illyrica Tabacaru, 1971. Most individuals were above the barcoding gap, meaning that for each individual the difference between the distance to the NN (Nearest Neighbour) and the distance to the furthest conspecific is above zero (Figs. 1B and 1C).

The mean intraspecific p-distance distribution is partially overlapping with distance to the nearest neighbour distribution (Fig. 1C), but for most species, nearest-neighbour distances were on average several times higher than maximum intraspecific distances (Fig. 1B). Maximum intraspecific p-distance was positively correlated with the number of individuals per species (Fig. 1D) (the correlation coefficient (ρ) = 0.447). For several neighbour species pairs, maximum intraspecific p-distance values were higher than their nearest-neighbour distance: Leuctra albida Kempny, 1899/Leuctra mortoni Kempny, 1899; Leuctra fusca (Linnaeus, 1758)/Leuctra albida; Perla pallida Guérin-Méneville, 1838/Perla marginata (Panzer, 1799) and Isoperla illyrica/Isoperla tripartita. For the following neighbour species pairs, p-distance to the nearest neighbour was below 2%: Leuctra albida/Leuctra mortoni, Nemoura cf. rivorum Ravizza & Ravizza Dematteis, 1995/Nemoura flexuosa Aubert, 1949, Perla illiesi Braasch & Joost, 1973/Perla burmeisteriana, Perla sp./Perla marginata, Perla pallida/Perla marginata and Isoperla illyrica/Isoperla tripartita.

This study resulted in five entities, which morphologically differ from their congeners and genetically appeared as separate lineages. These species therefore represent candidates for new, previously undescribed species: Leuctra cf. prima Kempny, 1899 (clade No. 7 in Fig. 2, distance to NN = 9.3%), Leuctra cf. inermis Kempny, 1899 (clade No. 4 in Fig. 2, distance to NN = 3.6%), Protonemura cf. autumnalis Raušer, 1956 (clade No. 72 in Fig. 2, distance to NN = 8.7%), Isoperla cf. lugens (clade No. 37 in Fig. 2, distance to NN = 6.7%) and Taeniopteryx n.sp. CRO-1 (clade No. 47 in Fig. 2, distance to NN = 7.1%) (Fig. 2, Table S4). Another separate lineage was recently described Isoperla popijaci Hlebec & Sivec, 2021 (clade No. 35 in Fig. 2, distance to NN = 6.7%) (Hlebec et al., 2021).

All methods used for phylogenetic reconstruction (ML (Fig. 2), NJ (Fig. S2) and BA (Fig. S3)) recovered the same, well-supported topology. All methods grouped phenotypically defined species in distinct, highly supported monophyletic species clusters (ultrafast bootstrap support >99). Phylogenetic relationships above the species level are in concordance with morphology-based hypotheses. All genera were monophyletic, while analyses resulted in unresolved phylogenetic relationships between families Capniidae and Taeniopterygidae.

Obtained sequences were allocated to 85 BINs (of which 29 were unique to BOLD), and delimited OTUs were mostly consistent with the clustering pattern observed in the ML tree, which was also concordant with morphological identification. Twenty-six BINs were represented by a single individual (singletons).

One BIN often belongs to a single species delineated by traditional taxonomy (Hausmann et al., 2013), and every different case can be an incentive for re-evaluation of morphological and molecular data (Hendrich et al., 2014). Specimens of several species showed deep COI divergence resulting in multiple BINs within a species: Protonemura praecox (Morton, 1894) (BOLD:AEH4111 and BOLD:AEH7722), Perlodes microcephalus (Pictet, 1833) (BOLD:AAL2343 and BOLD:AEH5507), Nemoura marginata Pictet, 1836 (BOLD:AAN1631, BOLD:AEH3564 and BOLD:AEK9273), Isoperla illyrica (BOLD:AEH3875 and BOLD:AEH7030), Leuctra fusca (BOLD:AAE6442 and BOLD:ACY3863), Leuctra albida (BOLD:AAM4011 and BOLD:AEH5504), Leuctra hippopus Kempny, 1899 (BOLD:ACL7184 and BOLD:AEH4770), Isoperla tripartita (BOLD:AEH3875, BOLD:AEH3876, BOLD:AEG6510 and BOLD:AEH7030), Isoperla grammatica (Poda, 1761) (BOLD:AEH6396 and BOLD: AEG4373) and Isoperla inermis (BOLD:ACS6073, BOLD:AAZ7905 and BOLD:AEH8653). Intraspecific p-distances were as follows for the following species: P. praecox (0–3.2%), P. microcephalus (0–4.2%), N. marginata (0–4.8%), Isoperla illyrica (0.002–6.8%), Leuctra fusca (0–5.8%), Isoperla tripartita (0–6.6%), Isoperla grammatica (0–5.0%) and Isoperla inermis (0–4%).

A shared BIN assignment was obtained within three genera: Isoperla, Perla and Nemoura, and species Isoperla illyrica/Isoperla tripartita, Nemoura flexuosa/Nemoura cf. rivorum and Perla pallida/Perla sp./Perla marginata.

Overall, use of different species delineation algorithms resulted in a different number of putative species. ASAP, with the best ASAP-score (6.00) which was achieved at a distance threshold of 2.5%, delimited 76 putative species and achieved a maximum match score with morphology (97%). The partition with the second-best ASAP-score (8.50, distance threshold 3.6%) delimits 70 species and the third partition (9.00, distance threshold 5.0%) delimits 64 species. The ABGD method delineated 62 putative species, while mPTP delineated 61 putative species. ABGD and mPTP represent conservative estimates of DNA species. Results of all species delineation methods are shown in Fig. 2.

Median joining (MJ) network depicted relatedness and distribution within the newly obtained divergent lineage (also unique to BOLD) named as Isoperla cf. lugens. The MJ network among 12 unique haplotypes which were separated by a different number of mutational steps is shown in Figs. 3A and 3B. The MJ network also revealed low haplotype sharing among sampling sites. Haplotypes CROPL311-21, CROPL344-21, CROPL127-21, CROPL310-21, CROPL312-21 and CROPL352-21 grouped together in a well-supported subclade (Fig. 2, Fig. 3B). Haplotype CROPL352-21 (discovered in Vitunjčica River) was separated by five mutational steps from the closest haplotype that was recorded in Drakulić rijeka, Crna rijeka and Čabranka River. The remaining seven haplotypes also grouped together in a well-supported subclade (Fig. 2, Fig. 3B).

Discussion

The present study represents the first comprehensive research combining morphological and molecular identification of stonefly species in Croatia and establishes DNA barcoding as an effective tool for reliable species identification. Such an approach enhances taxonomic resolution and assists in the quality of faunal research and can be used in discovering cryptic diversity and species complexes (Zangl et al., 2021). A taxonomically, well-curated, DNA barcode library can democratize species level work for non-experts with application in ecological research and in conservation biology across taxa, communities, and ecosystems. Other studies with specimens from Croatia that focus on mollusks (Buršić et al., 2021), mosquitoes (Bušić et al., 2021) and caddisflies (Kučinić et al., 2013; Valladolid et al., 2020) have obtained similar results, which support the efficacy of DNA barcoding for species determination.

So far, sequencing of the COI gene fragment has been used to elucidate the systematics and phylogeography of Plecoptera (Fochetti et al., 2009; Fochetti et al., 2011; Weiss, Stradner & Graf, 2011), and identify new species (Boumans & Murányi, 2014; Graf, Pauls & Vitecek, 2018; Pelingen & Freitag, 2020; South et al., 2019) as part of DNA barcoding initiatives (Morinière et al., 2017; Gattolliat et al., 2016; Ferreira et al., 2020) and revisionary systematics (Fochetti et al., 2011). Distinct DNA lineages obtained within morphospecies indicate the need for re-examination of morphological characters (Muster & Michalik, 2020; Wachter et al., 2015).

Within this study, all methods for phylogenetic reconstructions show that most species can be distinguished using COI barcodes. Exceptions are the following pairs: Isoperla illyrica/Isoperla tripartita, Perla pallida/Perla sp., Perla pallida/Perla marginata, Perla burmeisteriana/Perla illiesi, Leuctra albida/Leuctra mortoni, Leuctra albida/Leuctra fusca, Nemoura cf. rivorum/N. flexuosa, which possessed identical or overlapping COI sequences. For the above-mentioned species, identification is sometimes difficult due to high levels of intraspecific morphological variability between closely related species (Ravizza & Ravizza-Dematteis, 1995; Sivec & Stark, 2002; Murányi, 2011) and often both sexes and eggs are necessary for identification. BIN sharing reported within the species of the four genera: Isoperla, Perla, Leuctra and Nemoura potentially indicates introgression or hybridization.

High numbers of intraspecific BINs (obtained for ten species) can indicate discrete geographical populations of a species or overlooked cryptic species (Hawlitschek et al., 2017; Morinière et al., 2017), e.g., Dinocras cephalotes (Curtis, 1827) (Elbrecht et al., 2014), so the number of 85 BINs cannot be a proxy for the total number of stonefly species in Croatia. Nevertheless, the underlying RESL algorithm is based on a distance threshold of 2.2%, so it is expected to have a larger number of BINs in the dataset. Meier et al. (2021) made a claim that the cytochrome oxidase I (COI) barcode region cannot be used as the only/main data source for describing or delimiting species (Sharkey et al., 2021) and stressed the importance of additional species delimitation methods as well as examination of morphological characters for justifying the validity of a given species. COI barcode clusters (“BINs”) as a basis for species descriptions, given the assumption that a BIN equals a species, is often not consistent with the results of other species delineation methods (Meier et al., 2021) due to theoretical and empirical reasons (Puillandre, Brouillet & Achaz, 2021; Zhang et al., 2013).

Intra- and interspecific distances overlapped for some species (Fig. 1). Existence of a barcoding gap allows the use of DNA barcoding to identify species (Meyer & Paulay, 2005). Overlapping intra- and interspecific p-distances can be represented against a universal cut-off value (Collins & Cruickshank, 2013). Also, they can be a consequence of inaccurate taxonomy, indicating oversplit or cryptic species, especially for species within the genera Isoperla and Perla (Fig. 1).

In the present study, the distance to the NN is predominantly higher than the maximum intraspecific distance, confirming the clear local barcoding gap, enabling successful use of DNA barcoding for Croatian stonefly identification (Table S4). Furthermore, the congeneric average distance among species was five times higher than the average distance within species.

Nevertheless, the observed overlap between intra- and interspecific p-distances may be related to the presence of cryptic species or species complexes, which would not be surprising given the results of the morphological study.

Geographic morphological variation Genus Perlodes

Previous research of the stonefly fauna in Croatia, based exclusively on morphological analysis, have established the presence of specimens, which could not be identified with certainty to known species (Popijač & Sivec, 2009b). Perlodes intricatus (clade number 32, Fig. 2) from Plitvice Lakes, in which morphological differences were observed with respect to the typical Perlodes intricatus, was also found during this study. Molecular methods confirmed identification, whereas DNA barcoded specimens were grouped with sequences of P. intricatus retrieved from BOLD database, into a highly supported monophyletic clade (Dataset 1, Table S5), but with high intraspecific p-distances (5.2–5.6%), indicating the need for further field research across the entire range of the species. For another species, from the same genus, P. microcephalus, we obtained high intraspecific p-distance, and it is accompanied by increasing morphological variability from type specimens (Dataset 1, Table S5).

Genus Isoperla

The genus Isoperla is characterized by several poorly circumscribed West Palearctic species (Zwick, 2004) and Murányi (2011) argued for the need of taxonomic revision. Within Isoperla inermis (clade number 34, Fig. 2), we found great morphological variability as observed in previous research (Popijač & Sivec, 2009b). Individuals from Plitvice Lakes are almost double in size compared to specimens from Cetina River (Popijač & Sivec, 2009b), which may be a result of different climates and a longitudinal gradient. Colour variation is found in the abdomen, head, and pronotum, which vary from brown to black. Phylogenetic analysis of sequences from all Isoperla species and from I. difformis (Klapálek, 1909) from central Europe, resulted in a highly supported monophyletic clade I. inermis-I. difformis, which could ultimately result in the synonymy of these species (Dataset 2, Table S5).

Isoperla cf. lugens (clade No. 37, Fig. 2) was recorded in the area of the Plitvice Lakes (Popijač & Sivec, 2009b), and was determined based on similarity of the penial armature. During comprehensive field research in this study, specimens were found associated with several headwaters of karst rivers. The species differs morphologically from the alpine species, I. lugens, by having a lighter coloured head and pronotum and different penial armatures. In addition to these morphological characteristics, the species is also characterized by exceptional genetic distinctiveness. The lowest interspecific p-distance value from I. cf. lugens compared to other congeners from the I. tripartita and I. rivulorum species-groups is 6.7% and it represents a separate genetic lineage within the clade consisting of typical I. lugens and I. popijaci (clade No. 37, Fig. 2) (Dataset 3, Table S5). Due to the above-mentioned characteristics, Isoperla cf. lugens most probably represents a new species. The MJ network (Fig. 3B) for Isoperla cf. lugens revealed low haplotype sharing among sampling sites, which may be due to the small number of specimens per sampling site. Additional specimens would be useful to test this hypothesis.

Genus Protonemura

Protonemura hrabei (clade No. 71, Fig. 2) from the Cetina and Zrmanja Rivers emerges at the beginning of summer, mostly due to climatic conditions, in contrast populations from Central Europe, which emerge in autumn (Popijač & Sivec, 2009b). Molecular analysis of P. hrabei sequences and sequences of closely related Protonemura species, confirmed morphological identification and individuals from the Cetina River form a highly supported, monophyletic clade with Protonemura hrabei from Central Europe with an intraspecific p-distance of 2.5% (Dataset 4, Table S5).

Genus Taeniopteryx

In a comprehensive study of the genus Taeniopteryx in the framework of this study, morphological differences were identified among newly collected individuals of Taeniopteryx n.sp. CRO-1 (clade number 47, Fig. 2), Taeniopteryx hubaulti (Aubert, 1946) (clade number 46, Fig. 2) and Taeniopteryx auberti Kis & Sowa, 1964, as well as museum specimens in Croatia, Slovenia, Bosnia and Herzegovina, Montenegro and Germany. The morphological differences between Taeniopteryx n.sp. CRO-1 and other species are accompanied by genetic distinctiveness, making Taeniopteryx n.sp. CRO-1 a candidate for a new species (interspecific p-distances ranged from 7.8–9.5%) as suggested by Popijač & Sivec (2009b).

Morphological analysis of T. hubaulti confirmed the variability of the femoral thorn on the hind legs, present in some individuals but varying in size, even though this character was originally described as absent (Aubert, 1946). This morphological character should be clearly visible in T. auberti (Kis & Sowa, 1964). Genetic analyses of our sequences and those retrieved from BOLD and GenBank (Dataset 5, Table S5) resulted in the unclear taxonomic status of T. hubaulti and T. auberti, so there is the suspicion that T. hubaulti may be a junior synonym of T. auberti, and morphological variability is a consequence of geographical distribution. This is also similar for the species, T. stankovitchi Ikonomov, 1978 and T. schoenemundi (Mertens, 1923), for which it has already been pointed out that additional research is needed to clarify their distinction (Fochetti & Nicolai, 1996). As has been noted, the genus Taeniopteryx is, from the taxonomic point of view, complicated, and oftentimes only females show reliable characters, so the whole genus needs revision.

Genus Leuctra

Within the Leuctra inermis species-group, congruence of morphospecies concepts and phylogenetic relationships among taxa was not studied until 2017 (Vitecek et al., 2017b). Vitecek et al. (2017b) found that relationships among species remained unresolved, suggesting sister taxon relationships between morphologically similar species and potential subspecies-level diversity. Morphological variability was observed within geographically isolated populations and the same was confirmed by the present study. As many species from the L. inermis species-group have overlapping geographical ranges, appearance of morphological variability within several species is expected (Fochetti et al., 2011) and mitochondrial introgression has already been confirmed within the Leuctra species pair, L. fusca and L. digitata (Boumans & Tierno de Figueroa, 2016). Thus, it can be assumed that assessment of drumming call variations could be helpful in resolving taxonomic relationships within this species-group (Vitecek et al., 2017b).

Few individuals collected in this study appeared as a distinct lineage (Fig. 2), which possessed morphological features resembling already known species: Leuctra cf. inermis (CROPL130-21) collected in the Plitvice Lakes National Park, Leuctra sp. collected at Cetina River (clade number 17, Fig. 2), Leuctra sp. Z (CROPL248-21) collected at Žumberak Hills and L. cf. prima (CROPL282-21, CROPL325-21 and CROPL326-21) collected at Papuk Mountain and near Plitvice Lakes National Park. Due to the unavailability of specimens of L. carphatica Kis, 1966, only found in the Carphatian Mountains, Slovenia and Austria (Andrikovics & Murányi, 2001), the comparison of collected individuals with this species was omitted. Morphological differences were also observed among individuals of L. mortoni and L. fusca, which requires further research with a more comprehensive sampling, which would contribute to a more precise taxonomy of these species.

Dinaric Karst as a biodiversity hotspot

The Dinaric Karst system of the Dinaric Mountains, represents one of the most diverse European freshwater habitats in terms of biological, geological and hydrological interplay, including many available microhabitats, which has resulted in speciation and endemism (Bonacci, 2009). Considering the results of earlier studies, high genetic diversity could be a result of the specific habitat requirements and biological characteristics of individual taxa, which promoted speciation and played an important role in the genetic differentiation of freshwater taxa (Previšić et al., 2009, 2014b, 2014a; Klobučar et al., 2013; Jelić et al., 2016; Szivák et al., 2017; Lovrenčić et al., 2020). The same goes for taxa restricted to particular microhabitats (caves, pits, underground and intermittent rivers and streams) within the Dinaric Karst (Bilandžija et al., 2013; Bedek et al., 2019; Pavlek & Mammola, 2021), which in general, can be considered as refugia from which taxa re-colonise Europe following glacial periods (Hewitt, 2000), often showing a pattern “refugia within refugia” (Kryštufek et al., 2007; Ursenbacher et al., 2008; Previšić et al., 2009; Jug-Dujaković et al., 2020).

The diversity of the stonefly fauna in Croatia is a probable consequence of the different climatic conditions in a variety of regions (Continental, Alpine and Mediterranean), a substantial altitudinal gradient, and an immense number of protected habitats. Within this study, we identified hotspots of elevated stonefly species. The highest levels of species richness were primarily located in the northwest Dinarides (Fig. 4): the border rivers, Kupa and Čabranka; Plitvice Lakes National Park; the Cetina and Una Rivers; and Mt. Papuk where several localities have five or more species. Hotspots of species richness are coincident with protected areas (e.g., Plitvice Lakes National Park, Mt. Papuk, Mt. Medvednica), which is not surprising, given that these areas have suitable conditions and mostly include fast streams with high oxygen saturation. Current research shows a significant decrease in the number of species from the northern part of the Dinaric Karst (Gorski kotar and Lika) to the Cetina River and the headwater of the Una River, which is consistent with the biological features of stoneflies. Great species richness is also confirmed by Ridl et al. (2018), who recorded 7–18 species across different study sites (in total 31 species) in the Plitvice Lakes National Park. Furthermore, Popijač & Sivec (2010) found significant species richness (14) within the Cetina River. The Croatian fauna shows great species richness, not only for stoneflies but also for other aquatic insects (Ivković & Plant, 2015; Vilenica et al., 2015; Vilenica et al., 2016; Kučinić et al., 2017; Vilenica, Ternjej & Mihaljević, 2021). Still, many parts of the Dinaric Karst are not well studied, due to the size of area and inaccessibility of the habitats. Species richness patterns are likely incomplete, especially from the aspect of the stonefly fauna.

Figure 4 Geographical location of the studied stonefly species.

Colours of dots represent the species richness in each locality and size of dots is proportional with the number. Main map (B) is an enlarged framed area in the bottom left corner (A). Approximate border of the Dinaric Karst according to Gams (2004). Map is produced with Cartopy package 0.19 in Python with use of elevation data from European Union, Copernicus Land Monitoring Service (2016).

Database’s enrichment and systematic implications

This study provides the first molecular characterization of nine species: Brachyptera tristis, Perlodes dispar, Leuctra bronislawi Sowa, 1970, Isoperla bosnica (Aubert, 1964), Isoperla illyrica, Isoperla albanica (Aubert, 1964), Perla carantana Sivec & Graf, 2002, Perla illiesi and Agnetina elegantula (Klapálek, 1905).

Genus Protonemura

Our molecular characterization of Protonemura auberti (Illies, 1954), P. hrabei, P. intricata (Ris, 1902), P. nitida (Pictet, 1836) and P. praecox should be considered in future revision of the genus, a need that has already been emphasized (Wagner et al., 2011; Vinçon, Reding & Ravizza, 2021). The discovery of specimens from the P. auberti species subgroup (determined as P. cf. autumnalis, clade number 72, Fig. 2) at the Plitvice Lakes National Park with morphological characteristics similar to P. aestiva Kis, 1965, which emerges throughout the spring and seems to be restricted to the Carpathian Mountains (Kis, 1974; Graf et al., 2009), suggests a hybrid of these two species, with range expansion of P. aestiva (Vinçon, Reding & Ravizza, 2021). Nevertheless, the production of hybrids with intermediate morphological characters has already been observed in the newly described P. bispina Vinçon, Ravizza & Reding, 2021, which is often vicariant with specimens of P. auberti (Vinçon, Reding & Ravizza, 2021). Therefore, it is necessary to pay additional attention to the genus also from that point of view. DNA barcoded specimens of P. auberti (CROPL257-21, CROPL258-21, CROPL281-21) collected during the spring at the Una River appeared as a separate clade among other specimens of P. auberti (clade number 74, Fig. 2). Establishing phylogenetic relationships with a multi-gene approach is necessary to unravel the taxonomy of this group.

Genus Dinocras

DNA barcoding of the species Dinocras megacephala (Klapálek, 1907) (clade number 23, Fig. 2), the only Dinocras species in Croatia, which is widely distributed in its northern and central part, emphasizes the need for revision of the BOLD database due to the noticeable number of erroneous determinations of this species. It often appears to be misidentified as D. cephalotes. Males of D. cephalotes differ from D. megacephala by having patches of stronger sensilla basiconica on the ventral side of the abdomen. Furthermore, brachypterous males of D. megacephala also occur at higher elevations (Illies, 1966), so brachyptery cannot be a characteristic to distinguish this species from D. cephalotes (usually at higher elevations).

Genus Zwicknia

Morphological examination of individuals from the Capnia bifrons Zhiltzova, 2001 species-group, following Murányi, Gamboa & Orci (2014) determined the presence of two species in Croatia: Zwicknia bifrons (clade No. 51, Fig. 2) and Zwicknia rupprechti (clade number 50, Fig. 2). Within the family Capniidae, we also discovered one of the smallest European species, Capnopsis schilleri balcanica (CROPL319-21) from only a single locality, a surprising find given that this was 17 years since the first finding (Murányi, 2004).

Genus Nemoura

Within the genus Nemoura, ten species were DNA barcoded (clades numbers 60–68, Fig. 2): N. avicularis Morton, 1894, N. cinerea (Retzius, 1783), N. dubitans Morton, 1894, N. marginata, N. minima Aubert, 1946, N. sciurus, N. flexuosa, N. cf. rivorum, N. mortoni (Ris, 1902) and N. uncinata Despax, 1934. The Nemoura flexuosa-marginata complex is one of the most enigmatic assemblages of species within European stoneflies and requires revision (Ravizza & Ravizza-Dematteis, 1995).

The flexuosa-marginata species-group is composed of widely distributed European species (N. flexuosa, N. marginata Pictet, 1836 and N. uncinata) and six Italian and French Alps endemic species: N. hesperiae Consiglio, 1960, N. lucana Nicolai & Fochetti, 1991, N. oropensis Ravizza & Ravizza Dematteis, 1980, N. pesarinii Ravizza & Ravizza Dematteis, 1979 (all occurring in Italy), N. palliventris Aubert, 1953 and N. rivorum Ravizza & Ravizza-Dematteis, 1995 (occurring in Italy and further north in the French Alps) (Fochetti & Vinçon, 2009). Descriptions of two species within that complex: Nemoura rivorum and Nemoura sabina Fochetti & Vinçon, 2009 helped in understanding morphological variation among individuals within this species complex. Nemoura rivorum, which is endemic to the northern section of the Apennines and exhibits a variable apical and arched sclerite of the epiproct, is often erroneously identified as N. flexuosa, especially when only females are available for morphological analysis. Pregenital plate shape is similar in all species belonging to the N. flexuosa-N. marginata species-group and separating nymphs among species is almost impossible. Specimens collected as part of this study (CROPL075-21, CROPL076-21 and CROPL162-21) have morphological characteristics similar to N. rivorum (determined as N. cf. rivorum). However, based on the similarity of the sequences, these specimens clustered with N. flexuosa (CROPL070-21, CROPL095-21, CROPL190-21), which further emphasizes the importance of species-group revision, and revision of the sequences in BOLD and GenBank databases, respectively.

Furthermore, high morphological variability has been observed even among specimens of Nemoura marginata Pictet, 1836 (clade number 64, Fig. 2) (Popijač & Sivec, 2009b). The mean intraspecific p-distance was 3.1%. Future research should assess whether N. marginata represents a single species with large intraspecific distances and high morphological variability or a species-complex, as stated earlier (Ravizza & Ravizza-Dematteis, 1995).

Genus Leuctra

This study also provides the first molecular characterization of Leuctra bronislawi (CROPL131-21 and CROPL132-21). This autumnal species, which is relatively rare and a relict species with disjunct distribution in the Balkan and the Carpathians, was recently found in the Czech Republic (Kroča, 2010) and the first early spring records were reported from the Republic of Macedonia (Murányi, Kovács & Orci, 2014). Considering the limited knowledge of the stonefly fauna in countries which can include potential distribution areas of L. bronislawi, it is anticipated that many more populations remain to be discovered and recorded.

Genus Isoperla

Within the mostly endemic genus Isoperla, three species were DNA barcoded for the first time: Isoperla bosnica (clade number 39, Fig. 2), Isoperla illyrica (clade number 42, Fig. 2) and Isoperla albanica (clade number 41, Fig. 2). Isoperla bosnica is, based on morphology, a member of the Isoperla oxylepis species-group, which was redescribed based on SEM studies of the penis and egg structure (Murányi, 2011). I. bosnica was hitherto reported only from type locality (SE Bosnia-Herzegovina), NW Macedonia and Montenegro (Murányi, 2011) and the medial penial armature, a basic diagnostic characteristic, is like the armature of Isoperla oxylepis (Murányi, 2011). Isoperla albanica has an Eastern Alpine-Illyrian distribution and is characterized by an undivided medial penial armature (Murányi, 2011). I. illyrica, described as an endemic species to the Postojna Cave entrance, is now common across a wide area of the Dinaric Karst. Our phylogenetic analysis does not resolve the placement of this species (Dataset 2, Table S5). A multi-gene approach across the entire range is a priority. A similar approach is warranted for the I. grammatica species-complex (Murányi et al., 2021).

Genus Perla

The taxonomy of Perla species is unresolved and constitutes a big challenge, and the most recent revision of the genus suggests using characters in the egg chorionic (Sivec & Stark, 2002) as reliable for species recognition. To revise such a problematic genus, the inclusion of genomic data is required. For some species, it is already considered that they represent a species-complex, such as P. pallida (distributed in the Caucasus, Anatolia, the Balkans, and the Carpathians) (Sivec & Stark, 2002), often erroneously identified as P. marginata. Furthermore, the taxonomic status of some species is uncertain, such as P. bipunctata Pictet, 1833 (Sivec & Stark, 2002) and specimens found within this study on Ruda River (marked as Perla sp., clade No. 29, Fig. 2), which morphologically differed from congeners but genetically represent one lineage and one BIN. Perla burmeisteriana was recorded for the first time in 1908 and a few larvae were found at the northern foot of Papuk Mountain (Popijač & Sivec, 2009a). Perla illiesi was found in several localities in Croatia (Kupa and Čabranka Rivers). Within the present study, this species was recorded at two localities in Lika and DNA barcoded for the first time (clade No. 26, Fig. 2), as well as Perla carantana (clade No. 24, Fig. 2), which was reported at several localities in Slovenia and Austria (Sivec & Graf, 2002).

Genus Besdolus

Interspecific p-distances between newly obtained sequences of B. imhoffi and B. illyricus retrieved from GenBank (10.6–12.4) do not support the synonymy of B. illyricus and B. imhoffi, as stated before (Fochetti et al., 2011) (Dataset 6, Table S5).

Extirpation and conservation

High sensitivity of stoneflies to abiotic changes may lead to local or global extirpation of taxa (Fochetti & Tierno de Figueroa, 2006; Graf et al., 2018). Extensive field research that began in Europe about twenty years ago, however, found the presence of taxa that were considered extinct, but with range reduction. The local and regional extinction rate of stoneflies would be the highest across the Animal Kingdom, according to IUCN criteria (Sánchez-Bayo & Wyckhuys, 2019). In neighbouring countries, such as Italy, several species can be considered already extirpated: Brachyptera trifasciata (Pictet, 1832), Isogenus nubecula Newman, 1833, Taeniopteryx nebulosa (Linnaeus, 1758) and Perla abdominalis Burmeister, 1839 (Fochetti et al., 1998; Fochetti, 2020), while even more species can be considered threatened with extinction. The situation is, as usual, also critical for endemic species (known only from their type locality or a few populations) and all microendemic taxa (Fochetti, 2020).

Despite the relatively large effort invested in researching the stonefly fauna in Croatia (Popijač, 2008; Popijač & Sivec, 2009a, 2009b), several species are preserved only in museum collections: Perla bipunctata, Perla grandis, Rambur, 1842 Isoperla obscura (Zetterstedt, 1840) and Isogenus nubecula (Popijač & Sivec, 2009a). It is questionable whether these species have gone locally extinct, or their populations have decreased so much that it is difficult to detect them. In future systematic research of the stonefly fauna, discovering these species and their distributions will be one of the priorities.

But in spite of that, we recorded many species that have been declared extirpated in other European countries. Brachyptera monilicornis (Pictet, 1841) has long been considered extinct because it does not occur in Central Europe (Zwick, 1992). In the border rivers of Croatia and Slovenia (Kupa and Čabranka), as well as in the streams and rivers of Papuk Mountain, B. monilicornis is very common. Furthermore, the rare and endangered lowland species, Taeniopteryx nebulosa and Rhabdiopteryx acuminata (Klapálek, 1905), have been recorded at several localities in the foothills of Papuk Mountain. Besdolus imhoffi was re-discovered in Croatia in 2005 (Plitvice Lakes National Park) (Kovács & Murányi, 2008) after a one-hundred-year-old record (Popijač & Sivec, 2009a). This finding was confirmed with molecular analysis in the present study at the Una and Cetina Rivers. Species of the genus, including B. imhoffi, have relictual distributions (Zwick & Weinzierl, 1995) and are sensitive to environmental perturbations (Fochetti et al., 2011). Marthamea vitripennis, a species lost from most of Europe (Zwick, 2004) due to destruction of river potamon, was found in the Rába River in Hungary (Kovács & Ambrus, 2000), but also in Croatia in 2011 (Popijač & Sivec, 2011), and during field research in this study in 2021, on the rapids of the Una River. Another rare plecopteran, found in the Rába River in Hungary (Kovács & Ambrus, 2000) is Agnetina elegantula, which we recorded during this study at Papuk Mountain.

Conclusions

The current study generated a validated national reference DNA barcode library for stoneflies in Croatia, which can support the implementation of cost-efficient DNA-based identifications and assessments to ecological status. DNA barcoding proved to be an effective tool for the identification and delimitation of some closely related species. Furthermore, this study provides several findings of species thought to be extirpated from Croatia and neighbouring regions, as well as the first molecular characterization of species with restricted distributions. For some genera (e.g., Isoperla, Taeniopteryx and Perla) an integrative revisionary examination based on more comprehensive geographic sampling and application of a multi-gene approach, especially on type specimens, is necessary for resolving taxonomic relationships. Identifying areas with high biodiversity, including both morphological variability and genetic diversity, will allow further protection of stoneflies and their habitats.

Supplemental Information

Supplemental Information 1 Macrophotographs of 26 reported stoneflies species.

1. Brachyptera seticornis; 2. Leuctra fusca; 3. Leuctra inermis; 4. Leuctra major; 5. Leuctra mortoni; 6. Leuctra nigra; 7. Leuctra prima; 8. Marthamea vitripennis; 9. Marthamea vitripennis; 10. Nemoura marginata; 11. Perla burmeisteriana; 12. Perla illiesi; 13. Perla marginata; 14. Protonemura hrabei; 15. Protonemura praecox; 16. Taeniopteryx n.sp. CRO-1; 17. Xanthoperla apicalis; 18. Brachyptera risi; 19. Brachyptera tristis; 20. Leuctra albida; 21. Nemoura avicularis; 22. Nemoura dubitans; 23. Nemoura sciurus; 24. Nemurella picteti; 25. Protonomeura nitida; 26. Zwicknia bifrons. Photographs taken by I. Sivec.

Click here for additional data file.

Supplemental Information 2 Alignment of obtained COI sequences.

Click here for additional data file.

Supplemental Information 3 Alignment of haplotypes used for median joining (MJ) network.

Click here for additional data file.

Supplemental Information 4 Sampling localities of stoneflies in Croatia and Slovenia.

Details about sampling sites are provided in Table S1. Symbols used on the map: black dots represent localities in Croatia and blue dots additional localities in Slovenia. Main map (B) is an enlarged framed area in the bottom left corner (A). Map is produced with Cartopy package 0.19 in Python with use of elevation data from https://land.copernicus.eu/imagery-in-situ/eu-dem/eu-dem-v1.1.

Click here for additional data file.

Supplemental Information 5 Circular Neighbour joining (NJ) cladogram based on the 268 COI haplotypes of Croatian stoneflies.

Size of dots at nodes is proportional to the bootstrap support value. Terminal codes present BOLD IDs, as in Table S1. The tree was annotated in iTOL ver. 5 (Letunic & Bork, 2021) and finished in Adobe Illustrator.

Click here for additional data file.

Supplemental Information 6 Bayesian inference (BI) phylogeny based on the DNA barcoding region (5′ fragment part of the mitochondrial COI gene).

Size of dots at nodes is proportional to the Bayesian posterior probability (BPP) categories. Terminal codes present BOLD IDs, as in Table S1. The tree was annotated in FigTree ver. 1.4.3 (Rambaut, 2009) and iTOL ver. 5 (Letunic & Bork, 2021) and finished in Adobe Illustrator.

Click here for additional data file.

Supplemental Information 7 List of specimens used in the study.

The information comprises BOLD IDs, GenBank accession numbers, BINs, Collection inventory numbers (ID; CNHM–Croatian Natural History Museum; CPSH–Collection of Plecoptera Sivec & Hlebec), species clade numbers indicating clades correspond to those in Fig. 2, families, names of taxa after morphological examination, collection dates and sampling sites. Abbreviations: Leg., legit; Det., determinavit.

Click here for additional data file.

Supplemental Information 8 List of PCR primers and thermocycling conditions used in study.

Click here for additional data file.

Supplemental Information 9 Averages of inter- and intraspecific uncorrected p-distances based on mitochondrial cytochrome oxidase subunit I (COI) barcode region.

Values were obtained through complete deletion option. Intraspecific p-distances are outlined by the black line.

Click here for additional data file.

Supplemental Information 10 Average intraspecific p-distance and distance to nearest neighbour specimen.

Click here for additional data file.

Supplemental Information 11 Datasets used in additional phylogenetic analyses.

Additional phylogenetic analyses were performed in several cases: for the species with high level of intraspecific morphological variability and species which differ mainly in taxonomic features from the species described so far.

Click here for additional data file.

Many thanks to Dr. Nikola Tvrtković for help and support in sample collection and Dr. Stephanie Loria for English language revision. We also thank Dr. Edward DeWalt and two anonymous reviewers for helping us to improve the manuscript.

Additional Information and Declarations

Competing Interests

Author Contributions

Field Study Permissions

DNA Deposition

Data Availability

The authors declare that they have no competing interests.

Dora Hlebec conceived and designed the experiments, performed the experiments, analyzed the data, prepared figures and/or tables, authored or reviewed drafts of the paper, and approved the final draft.

Ignac Sivec conceived and designed the experiments, performed the experiments, analyzed the data, authored or reviewed drafts of the paper, and approved the final draft.

Martina Podnar conceived and designed the experiments, authored or reviewed drafts of the paper, and approved the final draft.

Mladen Kučinić conceived and designed the experiments, authored or reviewed drafts of the paper, and approved the final draft.

The following information was supplied relating to field study approvals (i.e., approving body and any reference numbers):

The specimen collection was conducted in concordance with the approval of the Ministry of Economy and Sustainable Development of the Republic of Croatia (UP/I-612-07/21-48/73).

The following information was supplied regarding the deposition of DNA sequences:

The COI sequences are available at GenBank: OK316149 to OK316486 and MW907977 to MW907993.

The data are also available in the Barcode of Life Data System (BOLD): http://dx.doi.org/10.5883/DS-CROPL.

The following information was supplied regarding data availability:

The data are available in the Supplemental Files.

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
