# Peer review of "DNA barcoding for biodiversity assessment: Croatian stoneflies (Insecta: Plecoptera)"

_PeerJ, doi:10.7717/peerj.13213_

## Round 0.1 · original submission · Major Revisions

Dear authors,

Your article has been reviewed by 3 peer reviewers. The reviewers have provided evaluations and made recommendations for revisions to your manuscript. Reviewer 1 proposed few comments on the interpretations. Both reviewers 2 and 3 have concerns about the language. In particular, reviewers 3 gave many helpful corrections on your manuscript. Please also see the attached file.

I invite you to respond to the reviewers' detailed comments and revise your manuscript. All three reviewers' comments need to be addressed before the manuscript can be accepted.

Thank you for submitting your manuscript to PeerJ and I look forward to receiving your revision.

Best,

Xiaotian Tang, PhD
Academic Editor, PeerJ
[email protected]

Reviewer 1 ·

Basic reporting

no comment

Experimental design

no comment

Validity of the findings

no comment

Additional comments

This manuscript generated a validated national reference DNA barcode library for stoneflies in Croatia, which can support the implementation of cost-efficient DNA-based identifications and assessments to ecological status. The study will contribute to knowledge about the distribution of species, genetic lineages within species and systematic and phylogenetic relationships. This article appears to use standard methods for analyzing and presenting the data. The scientifically sound and the outcomes are relatively well explained. Therefore, I recommend its acceptance for publication in PeerJ after minor revision:
1) Line 90: The references should be carefully check, delete D and I in “D Hlebec & I Sivec, 2021”. The same errors also existed in Line 224, Line 394, etc.
2) Line 108: Popijač, 2008; Popijač & Sivec, 2009a, should be spelt in italic.
3) Line 223: Delete the last dot of “MrBayes 3.2.7.”.
4) Line 245: change ‘Taeniopteryx sp.’ to ‘Taeniopteryx sp.’. The same errors also existed in Line 390, Line 524, etc. Please check carefully.
5) Line 356: In the sentence ‘distance threshold 3,6%’, it should be 3.6%.

Reviewer 2 ·

Basic reporting

no comment

Experimental design

no comment

Validity of the findings

no comment

Additional comments

The manuscript is an important faunistic and genetic contribution. The authors present a comprehensive and taxonomically DNA barcode database for Croatian stoneflies, with the sufficient datas and analysis.

Comments and proposed corrections:

• The language of the manuscript is in need of a general check of a native speaker. As being not a native speaker, I avoid of making linguistic comments or corrections.

• Regarding Fig. 2: Almost all data are consistent with the family status, but Capnidae is split into two parts by the Taeniopterygidae dataset (in Fig. 2: 48, 50, 51). I suggest the authors to discuss this situation further.

• "Morphological variability" was repeatedly mentioned in the manuscript, and the attachment (peerj-69224-Supplemental_File_S1) also includes multiple photos of stoneflies. What puzzles me is that these beautiful images don't seem to present any morphological variability mentioned, or even contain any figures of taxonomic diagnosis (male terminalia , etc.). In addition, in these hundreds of datas, one species can correspond to several genetic sequences. So I wonder, do these sequences severally come from, and match the different morphotypes in each species? Is there some correlation between the morphological variability and their corresponding genetic diversity? Perhaps the authors can discuss more in this regard.

·

Basic reporting

Wow, this was a tremendous amount of work. I commend you on this ambitious undertaking. You have created a resource that will be used far into the future and you have built a professional roadmap for yourself and others to improve our understanding of the taxonomy and conservation ecology of one the most environmentally sensitive of aquatic groups in the world. Your approach is the route forward for Plecoptera taxonomy and systematics, and your work has application to conservation of aquatic systems as well. I look forward to great work from you.

The English in places is a bit rough. I have tried to suggest improvements. There are some places that I said I did not understand and state so. I think your prose would benefit from writing in the in the first person as opposed to passive voice.

You have covered the literature well, though I have suggested some North American literature that is pertinent.
The article is professionally done. The analyses are thorough. It is a totally original work.

Experimental design

Well done throughout. This is how I want to see taxonomists enhance their good work with molecular approaches.

Questions are well-defined. This work certainly fills a gap in the knowledge of the distribution and genetic variation of species and contributes a roadmap for improving the taxonomy of one of the most environmentally sensitive aquatic groups in the world.

Methods are well documented and should be a model for biodiversity work.

Validity of the findings

Findings are impactful and novel. The work challenges the field of Plecoptera taxonomy to improve in the use of molecular data to circumscribe species and test hypotheses of distinctness of species.

The underlying data have been provided in myriad ways. Analyses are sound and varied in order to test for consensus.

Conclusions are sound and support results.

Additional comments

Some improvement in English is necessary but should not be difficult. I always markup manuscripts to the extreme. Use what you can.

In the results section, I had some difficulty understanding what you were trying to do. I thought you might want to separate the morphological from the molecular results somewhat. I think you might want a Table 1 of species found that is not supplementary. I had some difficulty with your counts of species, so I looked at your specimens supplementary table and could get the numbers close to what you state in the results. It was not until I looked closely at Figure 2 that I saw exactly where 74 species came. You did a great job explaining that circular gene tree.

I don't know what the journal wants, but you don't usually provide author and year with species names, but in some places you did. Be consistent.

You present in some place proportions for genetic difference, others %, but the figures are in %. I expected they would be the same. % is easier to understand in text.

Figures 1 B & C are not particularly informative. The horizontal scale is hardly used to accommodate the data. 0-10% seems to be the appropriate data range . I also thought they might be better as histograms.
Figure 4. Some dots on this map are larger than others. You have richness color-coded, but there is no legend for size of dots.

You frequently talk about "obtained sequences". I think you try to distinguish between your sequences and those from BOLD or GenBank. Maybe here is a good place to use active, first person to shorten the sentence and be more direct, e.g., "Our sequences".

The Discussion is a little long. They seem to be a few to many asides in the text. Stick to the most important information in an effort to shorten the section. Also, in this section you begin a lot of sentences with an abbreviated genus name. I don't think that it is good practice to begin a sentence with an abbreviation and I try not to do it with an acronym either.

In relation to species you occasionally write "source areas" in relation to habitat. I don't know what you mean. Maybe spring sources or headwaters is you intent. If so, say that.

A few citation practices bothered me a bit. First, you citing my Plecoptera Species File for narrow distribution of a species or group is not appropriate and it robs your colleagues of a citation of their work that is specific to your needs. You also cite unpublished data a bit too much. Cite it when it is published--I expect you will have plenty of papers in the future. Also, you need to look at references that are broader than Europe. I have provided several North American references that should be specific to your needs.

Line 618. Species occur at elevations, not altitudes.
Line 692. Perla burmeisteriana Claassen, 1936 is the valid name for the homonym Perla abdominalis Burmeister, 1839. Perla abdominalis Guérin-Méneville, 1838 is also valid, the first instance of Perla abdominalis, by one year.

Throughout your subsection "Extinction and conservation" you use the term extinction to mean the global and local loss of a species. Please use the term extirpation for local loss or range loss.

Supplemental Table S1 has a few errors or suggested changes needed.
I would like to see a column for Family added. It will help others more effectively sort the data.
You record two variants of Leuctra hippopus, most with 1899 year of description, but one incorrect one with 1900.
You list a Leuctra sp. Z, is this correct? It appear nowhere else.
Isoperla popijaci does not have authority or year of description.
Taeniopteryx sp. has not temporary designation. I think that might be useful to you in the future. All are from Croatia currently. Maybe name it "Taeniopteryx n.sp.CR-1"

---

## Round 0.2 · accepted · Accept

Dear Authors,

Thank you for adequately addressing all the concerns raised during the initial review. I am pleased to inform you that your article, "DNA barcoding for biodiversity assessment: Croatian stoneflies (Insecta, Plecoptera)", has now been accepted for publication in PeerJ. Congratulations!

This is a great study. I really enjoyed handling and reading the manuscript. Thank you for your submission and we hope you will continue to support PeerJ.

Best,

Xiaotian Tang
Academic Editor, PeerJ
[email protected]